# G-Verifier: Geometric Verifier for Robust 3D Point Cloud Semantic Search with Spatial Relation Reasoning

## Abstract

Semantic search in *3D* point clouds is a fundamental task for *Spatial Intelligence* and embodied *AI*, yet it becomes particularly challenging when queries involve precise spatial relationships and current large-scale vision-language models often falter in these scenarios. Their reliance on monolithic, implicit attention mechanisms struggles to disentangle semantic attributes match from complex spatial geometric constraints, leading to unreliable localization. To address this issue, we introduce *G-Verifier*, a geometric verification module that enhances existing *3DVG* frameworks by explicitly decoupling the semantic attributes match and spatial reasoning processes. Our approach realizes a *Propose, Select, then Verify* paradigm, where *G-Verifier* acts as a post-hoc re-ranker, adjudicating semantically-filtered candidates based on explicit geometric facts. The core of our module is the *Rotary Spatial-Relationship Embedding (RoSE)*, a structured representation that dynamically fuses high-level object semantics with an explicit *3D* geometric encoding. We train this module using a specialized language-alignment strategy on our new large-scale dataset, *3D-SpAn*, which contains 285,177 structured spatial relationship annotations. Experiments on a challenging, manually-verified benchmark demonstrate the effectiveness of our approach. Our module itself achieves high *F1*-score(0.96) on a relational understanding proxy task, validating its strong discriminative power. When integrated into the end-to-end pipeline, *G-Verifier* improves grounding accuracy, increasing Acc@0.50(+2.50% over a strong baseline. Our work validates that a decoupled verification approach is a promising direction for improving the geometric reasoning capabilities of large-scale *3D* vision-language models.

## 1 Introduction

*Spatial Intelligence* is a cornerstone of autonomous systems such as robotics (Chen et al. (2024b)) and augmented reality (Baruch et al. (2021)), allowing them to understand and interact with the physical world. A critical aspect of this intelligence is the ability to interpret raw *3D* sensor data (Cai Y (2023)). Among the most common and informative *3D* representations are point clouds, typically captured by *LiDAR* and other depth sensors (Qi et al. (2017)). Each point corresponds to a spatial location $(x, y, z)$ on and is often augmented with attributes such as *RGB* color, intensity, and semantic labels. Collectively, millions of such points form a detailed geometric scaffold of a real-world environment, from a single object to an entire scene (Ling et al. (2023)), enabling machines to perceive environments in their native *3D* structure. Within a point cloud, distinct objects manifest as dense clusters of points (Sarker et al. (2024)), and their spatial relationships are implicitly encoded by the relative distances and orientations between these clusters (Shen et al. (2023)).

A key task for activating spatial intelligence is the *Semantic Search* of point clouds. The main idea behind this task is to identify a specific object within a point cloud scene based on a language description (Liu et al. (2024)). In computer vision, this task is widely known as *3D Visual Grounding (3DVG)*. The ability to perform *3DVG* with high precision is vital for a range of impactful applications. For instance, a service robot in a hospital must be able to unambiguously execute commands like "bring me the medical chart on the counter, not the one on the cart". Similarly, an augmented reality system for industrial maintenance needs to correctly highlight "the pressure valve located be-

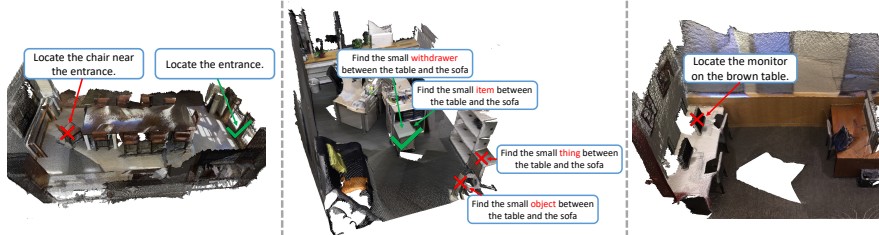

a) Geometric Ambiguity      b) Lexical Over-reliance      c) Semantic-Geometric Conflict

Figure 1: Common Issues for *3DVG* Models on Implicit Reasoning

hind the main pump" for an engineer (Attaran & Celik (2023)). Success in these applications hinges on a robust understanding of spatial relationships (Chen et al. (2022); Wang et al. (2023)).

While current models show remarkable progress (Chen et al. (2020); Achlioptas et al. (2020)), their ability to reason spatial relationships remains a significant bottleneck. Existing methods perform well when handling object-description semantics. However, once spatiotemporal semantics are introduced, the accuracy of object queries drops significantly if the descriptions are intricate, ambiguous, or conflict with the underlying scene geometry. (Xu et al. (2024b)). As illustrated in Figure 1, challenging cases for state-of-the-art *3DVG* models reveal several systematic failure modes:

*Geometric Ambiguity* arises when spatial constraints are underweighted in the final decision: a model can correctly locate the functionally defined `entrance` but still fail to `ground the chair near the entrance`, as its choice is dominated by visual similarity rather than precise relational reasoning.

*Lexical Over-reliance*. Grounding queries such as `the small withdrawer` based on specific keywords but failing when the same query is rephrased with more general terms like `item` or `object`, exposing a lack of compositional understanding.

*Semantic–geometric Conflict* occurs when the query specifies attributes inconsistent with the scene. For example, the model is asked to `find a monitor on the brown table`. Faced with a conflict (the only monitor is actually on a white table), it ignores the term of object attribute `brown` and hallucinates an incorrect grounding, failing to report the non-existence of the object.

Collectively, these failures highlight a core limitation, i.e., current models rely on monolithic, implicit reasoning mechanisms that entangle semantic interpretation with geometric validation, preventing them from reliably handling complex spatiotemporal semantics and undermining their robustness in human–robot interaction.

The current methodologies for spatial reasoning in *3DVG* can be broadly categorized into two paradigms in the following: 1) *Implicit Modeling*, where end-to-end foundation models are expected to learn spatial relationships as an emergent capability from massive data (Chen et al. (2024a); Cheng et al. (2024); Wu et al. (2024a); Chen et al. (2024c)); 2) *Explicit Modeling*, which seeks to inject more structure by either converting the scene into a symbolic representation like a *3D Scene Graph* (Armeni et al. (2019b); Hao et al. (2024)), or by directly embedding geometric features into the model's architecture (Ning et al. (2025)). Although distinct, both paradigms typically perform semantic and geometric evaluations in a coupled manner, which contributes to the fragility we observe.

Inspired by the success of re-ranking in *Information Retrieval (IR)* (Geigle et al. (2022)), we explore a third, emerging paradigm *Decoupled Verification*. The core insight is to separate the `what` (semantic understanding) from the `where` (geometric verification), moving beyond the limitations of both monolithic implicit models and coupled explicit approaches. We achieve this by proposing *G-Verifier*, a module that acts as an explicit geometric fact-checker. Instead of attempting to solve the entire problem in a single step, our framework first uses a powerful baseline model to select a pool of semantically plausible candidates. Then, *G-Verifier* combines its geometric verification with the baseline's semantic understanding to re-rank these candidates. This ensures the final selection is guided by spatial constraints while still respecting the strong semantic priors of the baseline model.

However, developing the geometric verification module is non-trivial, which includes several key challenges as follows.

***1) Spatial Relation Representation.*** Create a representation of abstract spatial relationships that is both expressive and compatible with object features. Our solution is the *Rotary Spatial-Relationship*

*Embedding*, a structured representation that dynamically fuses object-level features with a specialized *3D* geometric encoding derived from the objects' relative positions.

**2) Supervision Mechanism.** The training module is not effective. Existing *3D* grounding datasets lack the structured, explicit annotations of relational components (e.g., target, anchor, and relation type) required for our decoupled approach. To overcome this foundational data gap, we first constructed a large-scale dataset with 285,177 spatial relationship instances by fusing and improving existing resources. Instead of a naive classification objective, we devise a specialized language alignment strategy. This approach enables the module to learn a rich representation space for fine-grained similarity matching, which is crucial for its role as a verifier at inference time.

**3) Module Robustness.** It is challenging to ensure that the module remains stable and does not compromise performance in straightforward scenarios. We address this by designing a cautious re-ranking mechanism. It ensures geometric evidence only overrides the baseline's initial judgment when the spatial relationship is clear and unambiguous, thereby preserving high accuracy in simpler queries.

Our main contributions are summarized as follows:

- We propose a novel geometric verification module *G-Verifier* that facilitates a robust *propose, select, then verify* paradigm, effectively decoupling geometric validation from semantic selection.
- We design a novel *Rotary Spatial-Relationship Embedding* that fuses object semantics with explicit *3D* geometric information, enabling robust and fact-based re-ranking of candidates.
- We introduce a dedicated language-alignment strategy that enables *G-Verifier* to learn abstract spatial concepts from semantic targets.
- We construct and release a large-scale *3D* spatial relationship annotation set with 285,177 structured spatial relationship instances, providing a solid foundation for training and future research.

## 2 BACKGROUND

### 2.1 POINT CLOUDS AND 3D SCENES

Point cloud data is the primary modality for representing *3D* spatial environments, typically captured by sensors such as *LiDAR* or *RGB-D* cameras (Qi et al. (2017); Cai Y (2023)). It is a set of discrete points in a three-dimensional coordinate system, formally denoted as $P = \{p_i\}_{i=1}^{N}$, where each point $p_i \in \mathbb{R}^{3+d}$ includes its geometric coordinates $(x, y, z)$ and optionally $d$ additional attributes such as color or normal information.

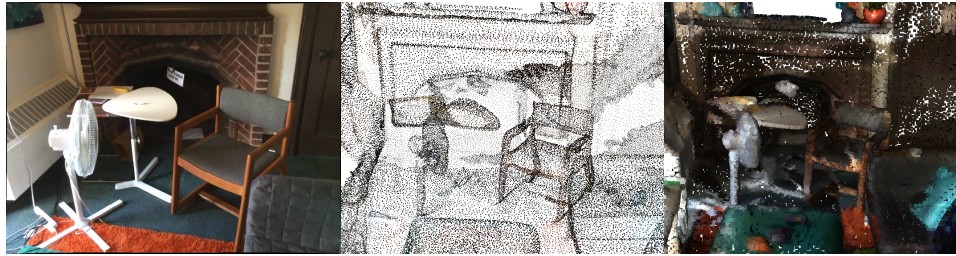

Figure 2: A Real-world *3D* Scene Represented by Point Clouds

Millions of such points form a detailed geometric scaffold of a real-world environment, from a single object to an entire scene (Dai et al. (2017); Armeni et al. (2016)). Within a scene, a physical object instance $o_k$ is represented by a subset of these points, $o_k \subset P$. The spatial relationships between objects are therefore implicitly encoded by the relative geometric arrangement of these point subsets.

Figure 6 illustrates how a real-world *3D* scene is represented as point cloud data. The leftmost panel shows an *RGB* image of an indoor environment. The middle panel depicts a sparse geometric point cloud, capturing the basic structure of the scene. The rightmost panel presents a dense point cloud with color attributes, providing a richer, more detailed representation for semantic analysis. This type of data forms the basis for tasks involving semantic search and spatial reasoning, as will be elaborated in the following sections.

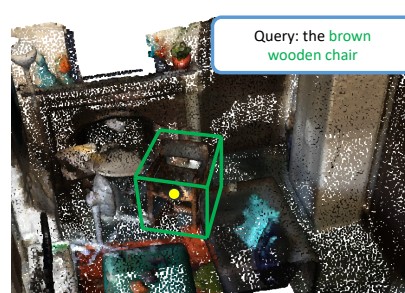 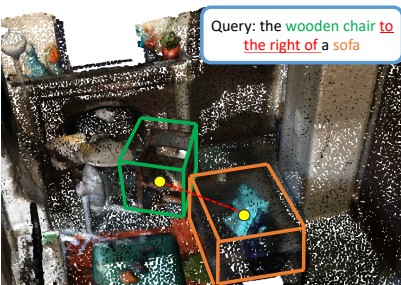

(a) Standard Semantic Search        (b) Spatial Semantic Search

Figure 3: Distinction between Standard Semantic Search and Spatial Semantic Search

## 2.2 SEMANTIC SEARCH IN 3D POINT CLOUD SCENES

**Definition 1 (Semantic Search)** *A semantic search query $Q_s$ is to identify a specific target object $o^*$ from the set of all objects $O = \{o_1, ..., o_k\}$ in a point cloud scene $P$, based on a natural language description L. (Liu et al. (2024))*

Semantic search, commonly referred to as *3DVG* in the computer vision field, has largely converged on a two-stage *Propose-then-Select* paradigm (He et al. (2021); Luo et al. (2022)). At first, a language-agnostic detector, such as *PointRCNN* (Shi et al. (2019)), generates a set of object proposals from the point cloud. Second, a language-guided selection module, often based on multimodal Transformers (Chen et al. (2020); Achlioptas et al. (2020)), matches the query $L$ against these proposals to identify the target object. This paradigm serves as the foundation upon which our work builds and extends.

Figure 3(a) depicts a standard semantic search that `find the brown wooden chair`. It relies solely on the target object's intrinsic attributes for localization. However, our work extends the semantic search by focusing more on the accuracy of spatial reasoning. For example, in Figure 3(b), the query `find the wooden chair to the right of a sofa` involves grounding a target object based on its spatial relationship to a distinct anchor object. We formally define *spatial semantic search* in the following section.

## 3 PROBLEM SETTING

While semantic search primarily targets object attributes, our focus is on queries where the final localization decision is critically dependent on understanding the objects' spatial relationships.

**Definition 2 (Spatial Semantic Search)** *Let a 3D scene $P$ containing a set of distinct object instances $O = \{o_1, o_2, ..., o_k\}$. A query $Q_{s^2}$ for semantic search with spatial reasoning is a natural language description $L$ that can often be deconstructed into a semantic triplet $L = \{O_t, R, O_a\}$. In this triplet, $O_t$ is the description of a target object, $O_a$ describes one or more anchor object(s), and $R$ represents the spatial relation that must hold between them.*

In this work, we scope the spatial relation $R$ to a comprehensive set of 12 fundamental types. We argue that this set is representative of the majority of spatial language used in human-centric scene descriptions. It is designed to cover a wide spectrum of cognitive spatial primitives, including: (1) projective relations that are dependent on a viewpoint (e.g., `left of`, `behind`), (2) topological relations describing contact or containment (e.g., `on`, `inside`), (3) a general proximity relation, and (4) more complex relations involving multiple objects (e.g., `between`). The full list of relations we address is: `left of`, `right of`, `front of`, `behind`, `above`, `on`, `under`, `below`, `inside`, `between`, `surrounded by`, and `proximity`. This categorization is consistent with large-scale empirical findings on human-generated descriptions in datasets like Visual Genome (Krishna et al. (2017)).

The main objective is to find a function $f$, that maps the scene and a query $Q_{s^2}$ to the correct target object instance: $o_t^* = f(P, L)$, where $o_t^* \in O$ is the unique instance that satisfies the composite constraints defined by the triplet $\{O_t, R, O_a\}$.

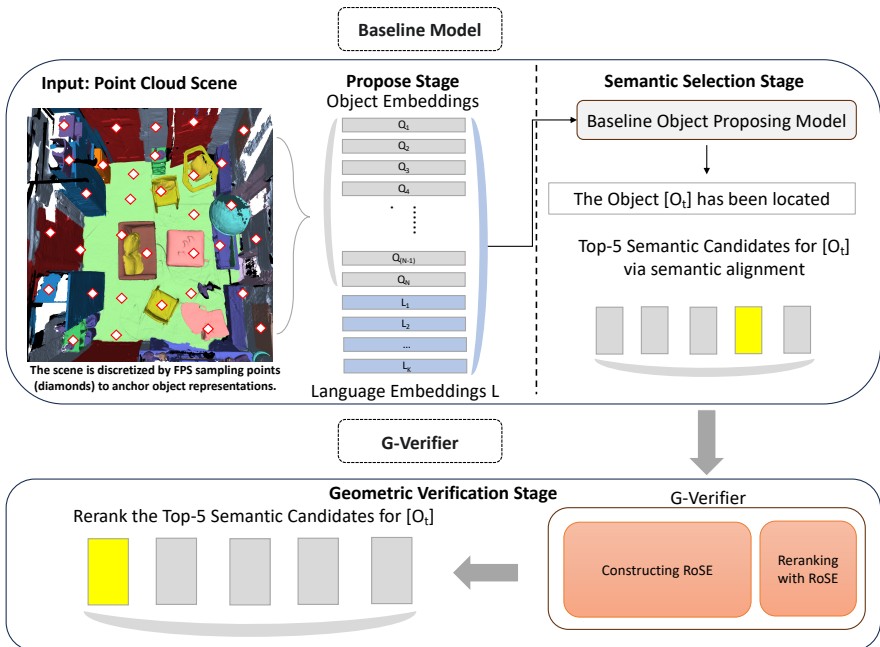

Figure 4: An overview of our *Propose, Select, then Verify* framework.

The success of this task is typically measured by the *Intersection-over-Union (IoU)* between the predicted *3D* bounding box of $o_t^*$ and that of the ground truth. This formulation is general; existing monolithic models attempt to learn the function $f$ as a single, end-to-end neural network, which often leads to a fragile coupling of semantic and geometric reasoning.

# 4  GEOMETRIC VERIFICATION MODULE

To address the fragility of existing 3D semantic search models in spatial reasoning, we introduce *G-Verifier*, an explicit geometric verification module. Our approach is built upon a decoupled *Propose, Select, then Verify* paradigm, designed to augment powerful, pre-trained foundation models. This section first presents the overall framework of this paradigm, then details the baseline implementation of the first two stages, and finally elaborates on the architecture, training, and inference mechanism of *G-Verifier* module.

## 4.1  OVERALL FRAMEWORK

Our method deconstructs the complex task of spatial relation grounding into a three-stage pipeline, as illustrated in Figure 4. The process starts with a standard baseline model, which performs the first two stages. In the propose stage, a 3D detector generates a set of class-agnostic instance embeddings from the input point cloud scene. In the semantic selection stage, an enhanced grounded *3D LLM* parses the language query $L$ to identify the textual descriptions for both the target ($O_t$) and anchor ($O_a$) objects. It then grounds these descriptions in the scene, producing ranked lists of *Top-K* semantic candidates for both the target object $C_t$, and the anchor object $C_a$. Our module then re-evaluates $C_t$ and $C_a$ through its internal components based on the query's spatial constraints $R$. Using a heuristic scoring mechanism that combines the baseline's semantic confidence with its own geometric verification score, *G-Verifier* produces a final, geometrically consistent re-ranked list of target candidates $C_t'$. The top candidate from this list, where $c_{t1}' \in C_t'$, is then returned as the final grounded object $o_t^*$.

## 4.2  THE BASELINE PROPOSE-SELECT PIPELINE

Our framework leverages a state-of-the-art *Grounded 3D-LLM*(Chen et al. (2024c)), which follows a query-based *Propose-Select* paradigm. It first generates a set of object instance representations from the point cloud, and then uses an *LLM* to select the candidate that best matches a language query. A detailed description of this baseline is provided in Appendix A.3.

### 4.3 G-Verifier: Our Decoupled Verification Module

*G-Verifier* performs the final, explicit geometric verification stage. It takes the top-$k$ semantically-ranked candidates from the baseline and re-ranks them based on the spatial relationship $R$ in query.

#### 4.3.1 Rotary Spatial-Relationship Embedding (RoSE)

To achieve reliable geometric verification, a structured representation that can precisely capture the relationship between a target-anchor pair is required. Our approach is conceptually analogous to the powerful *retrieve-and-rerank* architectures common in *IR* Geigle et al. (2022), where an efficient retriever first fetches plausible candidates, and a powerful re-ranker then meticulously examines this small set to produce a high-precision ranking. *G-Verifier* functions as this specialist re-ranker, and the *Rotary Spatial-Relationship Embedding (RoSE)* is its core component.

RoSE is not a static feature but a dynamically constructed composite embedding designed to facilitate a deep, cross-modal fusion of a specific target-anchor pair with a relational constraint. As shown in Figure 5, a *RoSE* vector is composed of four key parts:

- **Target and Anchor Context:** The instance embeddings $(O_t, O_a)$ of the candidate pair, which provide rich, instance-level semantic and visual information.
- **Explicit 3D Geometric Encoding:** A high-dimensional feature vector, $F_{\text{geom}}$, derived from the candidates' relative position vector, $\Delta p$. We employ a *3D extension of Rotary Position Encoding (RoPE)*(Su (2021)) for this purpose, as its formulation provides a natural inductive bias for learning relational representations that are independent of the objects' absolute positions. The detailed mathematical formulation of our *3D RoPE* is provided in Appendix A.4.
- **Learnable Relation-Type Embedding:** A learnable embedding, $E_R$, that acts as a semantic anchor for each of the 12 predefined relation types (e.g., `above`, `next to`).

These four components are then fused into the final RoSE vector using a lightweight *MLP*, as detailed in the following section.

#### 4.3.2 Feature Fusion Strategy in RoSE

A crucial design choice is how to integrate the four distinct feature components including the target instance embedding $O_t$, the anchor instance embedding $O_a$, the geometric encoding $F_{\text{geom}}$, and the relation-type embedding $E_R$ into a single, cohesive *RoSE* vector. We employ a non-linear fusion approach using a *Multilayer Perceptron (MLP)* Haykin (1994) to learn the complex interactions between these semantic and geometric features.

The components are first concatenated and then processed by a two-layer *MLP*, which includes a *ReLU* non-linearity Agarap (2018) to model complex dependencies. This process can be formally expressed as:

$$RoSE = MLP_{fusion}\left(Concat(O_t, O_a, F_{geom}, E_R)\right) \tag{1}$$

This *MLP*-based strategy allows the model to learn a rich, non-linear mapping from the constituent features to the final relational representation. It provides the flexibility to model how the interpretation of geometric features might be modulated by the semantic context of the objects and the

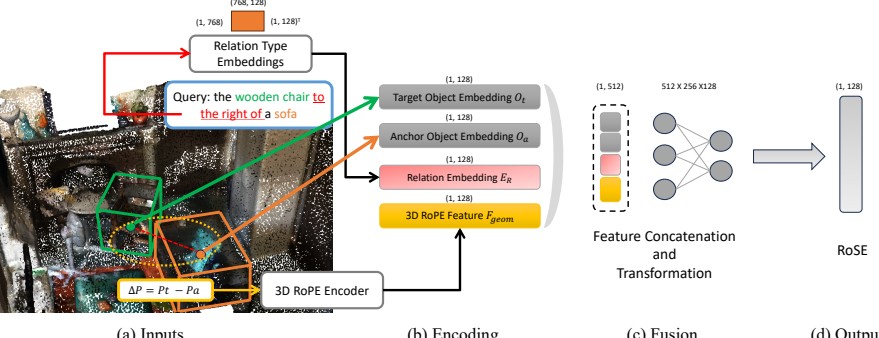

(a) Inputs    (b) Encoding    (c) Fusion    (d) Output

Figure 5: Architecture of constructing a single RoSE

high-level relational concept. A detailed breakdown of the *MLP* architecture is provided in Appendix A.9.

## 4.4 LEARNING BY DISCRIMINATING RoSE

The core challenge in training *G-Verifier* lies in learning the abstract concept of a spatial relation. A naive classification objective would suffer from a train-inference discrepancy. We therefore frame the training as a contrastive alignment task. Our goal is for our structured, geometry-infused *RoSE* representation to align with the *BERT* embedding of its corresponding natural language relation phrase $V_{\text{bert}}$.

To achieve this, we first construct a training set of positive triplets $(O_t, O_a, V_{\text{bert}})$. As our source datasets only provide ground truth for the target object, we devise an efficient 'inverse querying' pseudo-labeling strategy to automatically annotate the anchor object for over 296,000 relation instances. To create a discriminative learning signal, we then employ a structured hard negative mining strategy (*HSC Mining*), which generates plausible but incorrect configurations by perturbing the relation type or the object pair within the same scene. The detailed procedures for our pseudo-labeling and negative mining strategies are provided in Appendix A.5.

Finally, we train the *G-Verifier* using an in-batch contrastive loss, guided by an asymmetric semantic target (detailed in Appendix A.6). For each positive *RoSE* representation, $RoSE_i^+$, the objective is to maximize its similarity with its true language target, $V_{bert_i}^+$, while minimizing its similarity to all other language targets in the batch, which act as distractors. This is formalized for each positive sample $i$ as:

$$\mathcal{L}_i = -\log \frac{\exp(sim(RoSE_i^+, V_{bert_i}^+)/\tau)}{\sum_{j=1}^{B} \exp(sim(RoSE_i^+, V_{bert_j}^+)/\tau)} \tag{2}$$

where $\tau$ is a temperature hyperparameter. This training objective encourages the model to learn a rich and discriminative embedding space.

## 4.5 INFERENCE WITH G-VERIFIER

At inference time, *G-Verifier* functions as a re-ranking module that refines the initial candidate list from the baseline model based on an explicit geometric evaluation. The process consists of three main steps.

**1) Preparation of Inputs.** The process begins by parsing the user's language query $L$ into a structured triplet $\{O_t, R_{text}, O_a\}$ and encoding the relation phrase $R_{\text{text}}$ into a target vector, $V_{target}$, using our frozen BERT model. We then perform two separate queries to the baseline's semantic selection module with $O_t$ and $O_a$ to obtain the top-$k$ candidate lists $C_t$ and $C_a$, respectively.

**2) Geometric Verification and Scoring.** Our verification is constrained to the top-$k$ candidates in $C_t$ and $C_a$ to ensure computational efficiency. We iterate through all possible target-anchor pairs $(c_{ti}, c_{aj})$ from these lists, construct their *RoSE*, and compute a geometric verification score via cosine similarity with the language target:

$$Score_{\text{geom}}(i, j) = Sim(RoSE(i, j), V_{target}) \tag{3}$$

**3) Re-ranking via Weighted Score Fusion.** To effectively balance the baseline's semantic confidence with our geometric score, we employ a weighted fusion strategy. For each target candidate $c_{ti}$, we first compute its optimal, confidence-weighted geometric evidence, $Score_{\text{geom}}^{\text{best}}(c_{ti})$, by finding the anchor candidate that maximizes the product of the geometric score and the anchor's own semantic confidence. The final score is then a linear interpolation of the target's initial semantic score, $s_{ti}$, and this geometric evidence:

$$Score_{final}(c_{ti}) = (1 - \alpha) \cdot s_{ti} + \alpha \cdot Score_{geom}^{best}(c_{ti}) \tag{4}$$

where $\alpha$ is a balancing hyperparameter. The candidate with the highest final score is selected. This fusion strategy allows geometric evidence to act as a powerful refinement signal, while still respecting the baseline's robust semantic judgments. The detailed rationale for this fusion approach and our hyperparameter choices are provided in Appendix A.10.

## 5 EXPERIMENTS

To systematically evaluate our proposed *G-Verifier*, we conduct two sets of experiments. First, we perform a detailed component analysis to validate the design of our *RoSE* representation in an isolated setting. Second, we integrate *G-Verifier* as a post-hoc re-ranking module with a strong baseline to assess its impact on the end-to-end spatial reasoning task. All experiments are conducted on our newly constructed *3D-SpAn* dataset. A detailed description of *3D-SpAn* is provided in Appendix A.12 and the implementation details of experiment environment and a full description of our evaluation metrics are provided in Appendix A.13.

### 5.1 ANALYSIS OF THE RoSE DESIGN

To validate the design choices within *RoSE*, we conduct a progressive analysis by starting with a naive baseline and incrementally adding our key components. We evaluate these configurations on a proxy classification task, with fine-grained *F1*-score results summarized in Table 1.

The results of our component analysis clearly demonstrate a synergistic relationship between semantic and geometric priors. A naive baseline using only the *instance embeddings (IE)* achieves a 0.25 *F1*-score, confirming that pre-trained features contain some implicit spatial context, but it is insufficient for robust reasoning. Augmenting this baseline with our learnable *Relation Type Embedding ($E_{type}$)* causes a dramatic leap in performance to 0.95, establishing this high-level semantic prior as the most critical component for the task. While adding only the explicit *3D RoPE* encoding yields a more modest gain on its own, its crucial role is revealed in the full model. The final *RoSE*, which combines all three components, achieves the best performance across every relation category, with *RoPE* providing the most significant boosts on geometrically ambiguous relations like `left_of` and `right_of`. This progressive analysis validates our design, proving that the synergy of implicit context, a dominant semantic anchor, and an explicit geometric signal is essential for building a powerful relational representation. A more detailed discussion of these results is provided in Appendix A.15.

Table 1: Fine-grained *F1*-score analysis of our *RoSE* design across 12 spatial relation types. We progressively add relational priors to a naive baseline. The highest score in each column is highlighted. *Full Precision* and *Recall* results are available in Appendix A.8.

| Setup | *left_of* | *right_of* | *front_of* | *behind* | *above* | *on* | *under* | *below* | *inside* | *between* | *surrounded* | *proximity* | *Macro F1* | *W.Avg F1* |
|---|---|---|---|---|---|---|---|---|---|---|---|---|---|---|
| (1) Naive (*IE* Only) | 0.15 | 0.16 | 0.12 | 0.13 | 0.22 | 0.28 | 0.19 | 0.21 | 0.25 | 0.08 | 0.06 | 0.30 | 0.17 | 0.25 |
| (2) + *3D RoPE* (*IE + RoPE*) | 0.22 | 0.24 | 0.18 | 0.19 | 0.25 | 0.30 | 0.22 | 0.24 | 0.26 | 0.10 | 0.07 | 0.32 | 0.20 | 0.34 |
| (3) + $E_{type}$ (*IE + $E_{type}$*) | 0.82 | 0.81 | 0.79 | 0.78 | 0.94 | 0.96 | 0.92 | 0.93 | 0.90 | 0.75 | 0.70 | 0.97 | 0.80 | 0.95 |
| (4) Full (*RoSE*) | **0.88** | **0.87** | **0.85** | **0.84** | **0.95** | **0.97** | **0.93** | **0.94** | **0.91** | **0.81** | **0.75** | **0.98** | **0.85** | **0.96** |

### 5.2 END-TO-END RE-RANKING PERFORMANCE

To evaluate our method's real-world impact, we integrate *G-Verifier* as a post-hoc re-ranking module with a state-of-the-art *Grounded 3D-LLM*, using our own re-implementation as a controlled baseline. We conduct a comprehensive evaluation on a challenging, manually-verified 1,000-sample test set, employing both standard grounding metrics and a suite of *Information Retrieval* metrics.

**Overall Performance and Robustness Analysis.** Table 2 presents the primary results of our end-to-end evaluation. *G-Verifier* achieves an improvement on standard top-1 grounding metrics, boosting Acc@0.50 by a substantial +2.50% and Acc@0.25 by +2.60% over the strong baseline. We further analyze the module's behavior on two critical subsets of the data. The first is the *Top-1 Rectification Rate*, which measures how often *G-Verifier* can correct a failure by the baseline. For the 510 cases where the baseline's initial predictions with *IoU* below 0.25, our module successfully elevated a different candidate to the top-1 position with an *IoU* of 0.25 or higher in 6.27% of these instances. Achieving a Top-1 correction is a highly challenging task, as it requires not just a minor improvement

but a decisive re-ordering of the candidate list. The second diagnostic is the *Non-Deterioration Rate*, which assesses the module's safety. For the 490 cases where the baseline was already successful ($IoU \geq 0.25$), our module preserved a successful outcome in 98.78% of instances. These diagnostic results confirm that *G-Verifier* acts as a safe and effective enhancement, primarily targeting the baseline's failures while rarely disturbing its successes.

**Analysis of Ranking Quality and Performance by Relation Type.** To provide a more holistic assessment beyond Top-1 accuracy, we evaluate overall ranking quality using standard *IR* metrics. The results in Table 3a demonstrate a systematic improvement, with a significant gain in Mean Reciprocal Rank (MRR) confirming that the correct object is, on average, ranked higher. A fine-grained analysis, broken down by relation type in Table 3b, further reveals where G-Verifier's strengths lie. The most substantial improvements are seen in categories with objective geometric cues, such as on (+4.17% Acc@0.25) and `proximity` (+4.14%). These are precisely the types of topological and proximity-based relations our explicit geometric encoding is designed to excel at. In contrast, performance on more viewpoint-dependent relations like 'behind' is more modest. This indicates that our G-Verifier is highly effective at verifying objective geometric facts, while modeling more subjective, orientation-dependent relations remains a challenging frontier.

Table 2: Overall performance of *G-Verifier* on our curated 1,000-sample benchmark. We report standard grounding metrics (Acc@$k$) and key *IR* metrics (*MRR*, *NDCG@$k$*).

| Method | Grounding | | IR | | |
|---|---|---|---|---|---|
| | Acc@0.25 | Acc@0.50 | MRR | NDCG@1 | NDCG@3 |
| *Grounded 3D-LLM* | 49.00% | 46.30% | 0.5038 | 0.4720 | 0.5435 |
| *+ G-Verifier* | **51.60%** | **48.80%** | **0.5260** | **0.4984** | **0.5627** |
| **Improvement** | **+2.60%** | **+2.50%** | **+0.0222** | **+0.0264** | **+0.0192** |

Table 3: Detailed analysis of re-ranking performance on our benchmark. Evaluation with standard IR metrics and performance improvement (Acc@0.25) broken down by major relation types.

(a) *IR* Metrics.

| Metric | Baseline | Reranked | Improv. |
|---|---|---|---|
| *MRR* | 0.5038 | 0.5260 | +0.0222 |
| *MAP@1* | 0.3140 | 0.3410 | +0.0270 |
| *NDCG@1* | 0.4720 | 0.4984 | +0.0264 |
| *MAP@3* | 0.4327 | 0.4583 | +0.0257 |
| *NDCG@3* | 0.5435 | 0.5627 | +0.0192 |
| *MAP@5* | 0.4738 | 0.4964 | +0.0227 |
| *NDCG@5* | 0.6207 | 0.6348 | +0.0141 |

(b) Improvement by Relation Type.

| Relation | Samples | Baseline | Improv. |
|---|---|---|---|
| on | 216 | 56.48% | **+4.17%** |
| proximity | 266 | 43.98% | **+4.14%** |
| under | 26 | 53.85% | **+3.85%** |
| above | 32 | 53.12% | +3.12% |
| left_of | 133 | 44.36% | +2.26% |
| front_of | 104 | 51.92% | +0.96% |
| right_of | 151 | 48.34% | +0.00% |
| behind | 55 | 47.27% | +0.00% |
| surrounded_by | 10 | 40.00% | +0.00% |

## 6 CONCLUSION

This paper tackles the challenge of robust spatial reasoning in *3D* point clouds. We identified the entanglement of semantic and geometric reasoning in monolithic models as a core limitation and proposed *G-Verifier*, a novel geometric verification module built upon a decoupled *Propose, Select, then Verify* paradigm. The core of module is *RoSE*, a structured representation that fuses instance-level semantic context with an explicit, view-invariant geometric encoding. Experiments, conducted on a challenging, manually-verified benchmark, demonstrate that this decoupled approach is highly effective. *G-Verifier* significantly improves the top-1 grounding accuracy over a strong baseline, boosting Acc@0.50 by +2.50%. Furthermore, a holistic analysis with metrics confirms that our module systematically enhances the overall ranking quality of the baseline's predictions. This work validates that a decoupled geometric verification stage can effectively and safely complement the implicit reasoning of large multimodal models to resolve complex spatial reasoning queries.

## ETHICS STATEMENT

We adhere to the ICLR Code of Ethics. Our work focuses on foundational algorithmic advancements and we do not foresee direct negative societal consequences. We outline key ethical considerations below.

Our 3D-SpAn dataset is derived entirely from existing public academic datasets (e.g., ScanNet), and our process does not involve new data collection from human subjects. We have taken care to ensure our released annotations do not contain personally identifiable information.

We acknowledge that, like any AI technology for spatial understanding, our methods could potentially be applied in dual-use scenarios. Our research is intended for positive applications such as assistive robotics.

Our framework relies on large, pre-trained language and vision models, which may inherit societal biases from their training data. While a full audit is beyond this paper's scope, we acknowledge that mitigating such biases in 3D vision-language systems is an important area for future research.

## REPRODUCIBILITY STATEMENT

We have made a concerted effort to ensure the reproducibility of our work. All key components required for reproduction are detailed throughout the paper and its supplementary materials.

- **Source Code and Models:** We will release the full source code for our G-Verifier module, the data construction pipeline, and all evaluation scripts. Pre-trained weights for our G-Verifier will also be made available. The code will be hosted on an anonymous repository for review and will be transferred to a public GitHub repository upon publication.
- **Dataset:** Our newly constructed 3D-SpAn dataset, which forms the basis of our training, will be publicly released. We provide a detailed description of the data construction process, including the source datasets, the LLM-based parsing pipeline, and our pseudo-labeling strategy, in Section A.12 and a more detailed prompt in Appendix A.17.
- **Implementation Details:** We provide comprehensive implementation details for all experiments. The architecture of our G-Verifier and its core component, RoSE, are detailed in Section 4.3. Key hyperparameters for both the component analysis training and the final end-to-end re-ranking are provided in Appendix A.9.
- **Evaluation:** The evaluation protocol for our end-to-end experiments, including the construction of our manually-verified 1,000-sample benchmark and the use of Information Retrieval metrics, is described in Section 5.2. The code for reproducing all tables and figures in the paper will be included in our code release.

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

## A APPENDIX

### A.1 BACKGROUND ON 3D SEMANTIC SEARCH AND POINT CLOUD DATA

This section provides background on the core concepts and tasks relevant to our work.

**Point Clouds and 3D Scenes** A point cloud is the primary data modality for representing 3D spatial environments, typically captured by sensors like LiDAR or RGB-D cameras Qi et al. (2017). It is a set of vertices in a three-dimensional coordinate system, $P = \{p_i\}_{i=1}^{N}$, where each point $p_i \in \mathbb{R}^{3+d}$ includes its geometric coordinates $(x, y, z)$ and optional attributes. Millions of such points form a detailed geometric scaffold of a real-world environment, as illustrated in Figure 6. Within a scene, a physical object instance $o_k$ is represented by a subset of these points.

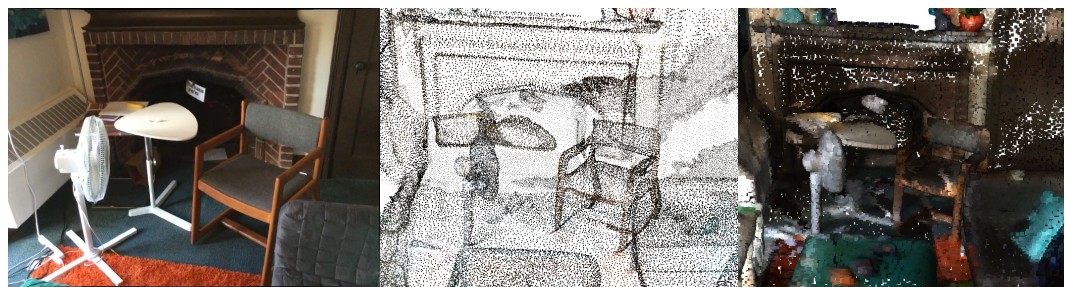

Figure 6: A real-world 3D scene represented as a point cloud. (a) RGB image. (b) Sparse geometric point cloud. (c) Dense point cloud with color.

**Standard Semantic Search (3D Visual Grounding)** A fundamental task that leverages this representation is semantic search, commonly known as 3D Visual Grounding (3DVG) in computer vision Liu et al. (2024). The goal is to identify a target object $o^*$ from all objects $O$ in a scene $P$ based on a language description $L$. The dominant approach follows a two-stage *Propose-then-Select* paradigm He et al. (2021); Luo et al. (2022), which serves as the foundation upon which our work builds.

### A.2 RELATED WORK

**3D Visual Grounding** 3D Visual Grounding (3DVG)(Liu et al. (2024)) seeks to link natural language queries to specific objects within a 3D scene. The field has largely transitioned from early methods relying on handcrafted features to end-to-end, Transformer-based paradigms. Pioneering works like ScanRefer(Chen et al. (2020)) and BUTD-DETR(Jain et al. (2022)) demonstrated the effectiveness of multimodal Transformers in implicitly learning the complex correspondences between visual features and textual phrases. The recent integration of Large Language Models (LLMs), as seen in Grounded 3D-LLM(Chen et al. (2024c)), has further advanced the state of the art by leveraging the powerful reasoning capabilities of LLMs for scene-level object localization through referent token mechanisms. A common characteristic of these powerful models is their reliance on implicit, attention-based mechanisms to handle spatial relationships, which motivates the exploration of more explicit modeling approaches.

**LLMs for 3D Scene Understanding**   The application of LLMs to 3D data is a rapidly advancing frontier. Research has progressed on two main fronts: object-level and scene-level understanding(Yang et al. (2023); Sharma (2023); Wu et al. (2024b)). At the object level, models like GPT4Point(Qi et al. (2024)) and PointLLM(Xu et al. (2024a)) have successfully enabled LLMs to comprehend the attributes and functions of individual point clouds. At the scene level, frameworks such as Grounded 3D-LLM(Chen et al. (2024c)) have made the crucial leap from single-object to multi-object reasoning. By providing all object proposals in a scene as context to an LLM, these models can resolve complex referential expressions. While significantly improving localization, this paradigm also highlights the remaining challenges in robustly handling inter-object spatial relationships, which is the primary focus of our work.

**Spatial Relationship Modeling in 3DVG**   To improve spatial reasoning in 3DVG, researchers have pursued several distinct strategies beyond relying solely on the model's implicit learning capabilities. One prominent direction is data-driven enhancement, where works like AugRefer(Wang et al. (2025)) programmatically augment 3D scenes and use Vision-Language Models (VLMs)(Chen et al. (2024a); remyxai (2024)) to generate new training data rich in spatial descriptions. Another approach involves imposing explicit structured priors, such as 3D Scene Graphs(Armeni et al. (2019a)). Methods in this category, like SGFN(Wu et al. (2021)), first generate a comprehensive graph of objects and their inter-relations, and then perform language-based reasoning on this pre-computed structure. A third major strategy is the direct injection of geometric features into the main reasoning pipeline. Classic examples include 3DVG-Transformer(Zhao et al. (2021)), which encodes relative distances and angles as additional input features, while more recent methods like ViewRefer(Guo et al. (2023)) transform these geometric priors into an attention bias to directly influence the scores within the Transformer(Dosovitskiy et al. (2021)). Our work presents a distinct alternative to these approaches. Instead of serving as an input feature, a pre-computed graph, or an attention bias, our G-Verifier functions as an independent, post-hoc verification and re-ranking stage with a new spatial relation representation *RoSE*. This architectural choice fundamentally separates the task of understanding objects from comprehending layouts, providing a more robust and interpretable pathway for resolving complex spatial constraints.

## A.3   The Details of our Baseline Propose-Select Pipeline

Our framework leverages a state-of-the-art 3D *Vision Language Model (VLM)* to execute the first two stages. While our *G-Verifier* is designed to be compatible with any model that can perform these foundational tasks. Here, we illustrate the specific baseline implementation used in our experiments.

*Object Proposal (Stage 1):* The initial stage generates object candidates from the input point cloud $P$. We follow the query-based detection paradigm popularized by *DETR* (Carion et al. (2020)). Specifically, our baseline employs a *Mask3D*-like (Schult et al. (2023)) architecture. It uses a set of $N$ learnable queries and a Transformer decoder, to predict a set of proposals $\{(m_i, O_i)\}_{i=1}^{N}$, where $m_i$ is a binary mask and $O_i$ is a high-dimensional feature representation for each instance, which we term the learnable instance embedding.

*Semantic Selection (Stage 2):* The second stage is a language-guided selection module. The core function is to take the set of all learnable instance embeddings and a textual description for an object (e.g., $O_t$) as input and to produce a ranked list of the top-$k$ candidates that best match this description. Many existing baseline methods can fulfill this role.

In our work, we employ a grounded *3D LLM*(Chen et al. (2024c)) for this purpose for its strong performance. This model implements the selection via a *generate-then-align* mechanism. It has been trained to generate a response containing a special referent token (e.g., `[ref]`), and its selection is made by matching the token's hidden state, $h_{\text{ref}}$, against all available learnable instance embeddings. This process is supervised by a grounding loss:

$$\mathcal{L}_{\text{grounding}} = \text{ContrastiveLoss}(h_{\text{ref}}, O_{gt}) \tag{5}$$

While powerful for grounding individual object descriptions, the end-to-end nature of this specific implementation couples semantic and spatial reasoning when faced with a compositional query $L$, which motivates our subsequent verification stage.

### A.4 Detailed Formulation of 3D Rotary Position Encoding

Our explicit geometric encoding is based on a 3D extension of Rotary Position Encoding (RoPE) Su (2021). RoPE was originally proposed for 1D sequences and encodes positional information by applying a rotation to feature vectors. A key property of RoPE is that the inner product between any two vectors at different positions depends only on their relative position, making it inherently suitable for relational modeling.

We extend this principle to 3D to encode the relative position vector between two object centers, $\Delta p = p_t - p_a = [\delta_x, \delta_y, \delta_z]$. The goal is to produce a $d$-dimensional feature vector, $F_{geom}$, where each component is a function of $\Delta p$. Each dimension $k \in \{0, 1, ..., d-1\}$ is computed as the sum of sinusoidal encodings from each coordinate difference $(\delta_x, \delta_y, \delta_z)$. These encodings are parameterized by frequencies $\theta_i = 10000^{-2i/d}$, where the frequency index $i$ is shared across pairs of dimensions, i.e., $i = \lfloor k/2 \rfloor$. The full formulation is as follows:

$$F_{geom}(\Delta p)_k = \begin{cases} \sin(\delta_x \cdot \theta_{\lfloor k/2 \rfloor}) + \sin(\delta_y \cdot \theta_{\lfloor k/2 \rfloor}) + \sin(\delta_z \cdot \theta_{\lfloor k/2 \rfloor}) & \text{if } k \text{ is even} \\ \cos(\delta_x \cdot \theta_{\lfloor k/2 \rfloor}) + \cos(\delta_y \cdot \theta_{\lfloor k/2 \rfloor}) + \cos(\delta_z \cdot \theta_{\lfloor k/2 \rfloor}) & \text{if } k \text{ is odd} \end{cases} \tag{6}$$

This formulation provides a continuous, high-dimensional representation of the relative geometry that is robust to small coordinate shifts and naturally encodes both distance and direction.

### A.5 Detailed Training Procedures

This section provides a detailed breakdown of the data construction and negative mining strategies used to train our G-Verifier.

#### A.5.1 Anchor Pseudo-Labeling via Inverse Querying

Our source datasets provide a ground truth mask only for the target object, not the anchor. To overcome this, we devise an efficient pseudo-labeling strategy for anchors based on 'inverse querying'. For a given relation instance $\{O_t, R, O_a\}$ and its ground truth target instance embedding $V_{gt}$, we proceed as follows:

1. **Positive Target Embedding:** The positive target embedding, $V_t$, is directly identified as $V_{gt}$.
2. **Inverse Query Construction:** We leverage the semantic symmetry of spatial prepositions. We define a new triplet $\{O'_t, R', O'_a\}$ where roles are swapped: the new target is the original anchor ($O'_t = O_a$), the new anchor is the original target ($O'_a = O_t$), and the relation is inverted ($R' = \text{inverse}(R)$, e.g., left of becomes right of).
3. **Anchor Pseudo-Labeling:** We feed this inverse query, specifically the new target description $O'_t$, to the baseline's semantic selection module. The Top-1 returned instance embedding is selected as our positive pseudo-label anchor, $V_a$.

We acknowledge this heuristic can introduce label noise, but our framework proves robust due to the scale of our dataset and the nature of contrastive learning.

#### A.5.2 HSC Mining: Hard Structural-Semantic Negative Mining.

To ensure the model learns a discriminative representation space, training with high-quality hard negatives is crucial. We employ an in-scene dynamic negative mining strategy, which we term HSC Mining, which generates challenging negatives by creating plausible but incorrect relational configurations using only objects present within the same scene. For a given positive training sample, defined by the instance embedding pair $(O_t, O_a)$ and the relation $R$, we construct negatives using one of the following strategies:

- Relation-type Substitution. We keep the instance embedding pair $(O_t, O_a)$ fixed but replace the ground truth relation $R$ with an incorrect relation type $R_{random}$ randomly sampled from our vocabulary (e.g., replacing above with in front of). This forces the model to learn the fine-grained semantic differences between relation types for the same geometric configuration.
- Antisymmetric Relation Swapping. For relations that have a clear antisymmetric counterpart, we construct a highly challenging negative by keeping the instance embedding pair $(O_t, O_a)$ fixed, but swapping the relation $R$ with its inverse, $R' = \text{inverse}(R)$ (e.g., left of becomes right of).

This compels the model to become sensitive to the precise directionality encoded in the geometric features.

- Object Pair Substitution. We keep the relation type $R$ fixed but replace the ground truth instance embedding pair $(O_t, O_a)$ with a different pair $(O_i, O_j)$. This new pair is formed by randomly sampling two different instance embeddings from the set of all available object proposals in the same scene. This teaches the model to ground the specific relation instance to the correct objects, rather than just recognizing that the relation type exists somewhere in the scene.

These strategies collectively generate a diverse and challenging set of negative samples, forcing the model to learn a robust and fine-grained understanding of how semantics, geometry, and specific object instances jointly define a spatial relationship.

### A.6 ASYMMETRIC DESIGN OF SEMANTIC TARGETS IN CONTRASTIVE ALIGNMENT

An important design choice in our training paradigm is the composition of the language-based semantic target, $V_{\text{bert}}$, used in our contrastive alignment loss. Specifically, for a relation triplet like `the chair next to the table`, we encode the phrase `next to the table` (relation + anchor) but deliberately exclude the target object's description (`the chair`). This asymmetric design might raise the question of why the anchor's description is included while the target's is not. This choice is fundamental to the effectiveness and learning objective of our G-Verifier module.

From a linguistic and cognitive standpoint, the anchor object is an integral part of the relational phrase itself. A prepositional phrase like `next to the table` defines a specific spatial region, and its meaning is incomplete without the anchor (`the table`) which acts as the reference or origin of the spatial coordinate system. The target object (`the chair`), in contrast, is the entity being described or located by this entire relational phrase. Therefore, by using BERT("`next to the table`"), we provide a pure, yet contextually rich, representation of the spatial relationship that we aim for our model to understand. The target vector encapsulates the semantics of a search space defined by the relation and the anchor.

More critically from a machine learning perspective, this asymmetric design prevents the model from learning a trivial shortcut and bypassing the core challenge of geometric reasoning. The Relational Query, $Q_R$, already contains the rich, instance-level visual features of the target object via its Object Query, $Q_t$. If we were to also include the target's textual description (e.g., `the chair`) in the BERT target vector, the model could potentially minimize the contrastive loss by learning a simple, unimodal mapping between the visual features in $Q_t$ and the text features corresponding to "the chair" in the BERT vector. This would create a path of least resistance where the model could achieve low loss by focusing on object matching, while neglecting the difficult but essential task of learning the geometric constraints encoded by our 3D RoPE module.

By providing only the relational phrase (relation + anchor) as the semantic target, we enforce a true multimodal alignment. The model is forced to learn how the visual-geometric configuration of an object pair, represented by the full Relational Query, corresponds to the linguistic description of their spatial relationship. This ensures that the G-Verifier genuinely learns to perform geometric verification, which is the central goal of our work.

### A.7 A PROBABILISTIC INTERPRETATION OF THE DECOUPLED PARADIGM

The distinction between our decoupled verification paradigm and monolithic end-to-end approaches can be further understood from a probabilistic modeling perspective. This section provides a formal interpretation of how these two paradigms differ in their approach to modeling the posterior probability of the target object.

Let $o_t^*$ be the correct target object instance, $V$ be the visual information from the 3D scene, and $L$ be the language query, which we deconstruct into its components $\{O_t, R, O_a\}$. The ultimate goal of any 3D visual grounding system for spatial reasoning is to find the object that maximizes the posterior probability $P(o_t^*|V, L)$.

**Monolithic Approach: Modeling a Joint Probability** State-of-the-art end-to-end models, such as the baseline Grounded 3D-LLM, implicitly attempt to directly model this complex, joint posterior

distribution. Their approach can be expressed as learning a single, high-dimensional function $f_\theta$ that maps the inputs directly to a score for each potential object $o_i$ in the scene:

$$P(o_t^* = o_i | V, L) \propto \exp\left(f_\theta(o_i, V, L)\right) \tag{7}$$

In this formulation, the semantic constraints from $O_t$ and $O_a$ are entangled with the geometric constraints from $R$. The function $f_\theta$, typically a large Transformer, must learn the intricate interactions between these different types of information from scratch within a single, unified computation. This tight coupling makes the model prone to failures when faced with novel or conflicting semantic-geometric configurations, as it lacks a structured way to adjudicate between different sources of evidence.

**Our Decoupled Approach: A Conditional Probability Decomposition**   Our *Propose, Select, then Verify* paradigm, in contrast, can be interpreted as a structured decomposition of the posterior probability, analogous to applying the chain rule of probability. Instead of modeling the complex joint distribution in one step, we break it down into a sequence of more manageable, specialized conditional probability estimations.

**Stage 1 & 2: Semantic Selection as Posterior Filtering.**   The first two stages, executed by the baseline model, can be viewed as estimating a semantic-only posterior probability, $P(o_t | V, O_t)$. This stage deliberately ignores the complex spatial relation $R$ and focuses only on identifying a set of candidates $C_t$ that are semantically plausible matches for the target description:

$$C_t = \{o_i \in O \mid P(o_i | V, O_t) > \epsilon\} \tag{8}$$

where $\epsilon$ is some confidence threshold. This step effectively uses semantic information to prune the search space.

**Stage 3: Geometric Verification as Conditional Re-ranking.**   Our G-Verifier then addresses a more constrained problem: estimating the probability of a candidate $o_t \in C_t$ being the correct target, now conditioned on the additional evidence provided by the spatial relation $R$ and the anchor $O_a$. This corresponds to modeling the conditional probability $P(o_t | V, L, o_t \in C_t)$.

Our Weighted Score Fusion mechanism is a practical implementation of this probabilistic reasoning. The final score can be interpreted as an approximation of the log-posterior, based on a Bayesian fusion of evidence. The log-posterior is approximated as a weighted sum of a semantic log-prior and a geometric log-likelihood:

$$\log P(o_t|V,L) \approx (1-\alpha) \underbrace{\log P(o_t|V,O_t)}_{\text{log-prior from semantics}} + \alpha \underbrace{\log P(o_t|V,R,O_a)}_{\text{log-likelihood from geometry}} \tag{9}$$

Our final scoring function,

$$\text{Score}_{\text{final}}(c_{ti}) = (1-\alpha) \cdot s_{ti} + \alpha \cdot \text{Score}_{\text{geom}}^{\text{best}}(c_{ti}) \tag{10}$$

implements this idea directly. The initial semantic score $s_{ti}$ from the baseline model serves as the semantic log-prior. Our computed best geometric score, $\text{Score}_{\text{geom}}^{\text{best}}$, acts as the geometric log-likelihood, representing the evidence for the relation holding true. The hyperparameter $\alpha$ balances the confidence between the prior belief from the semantic selector and the new evidence from our geometric verifier. By decomposing the complex joint probability into a weighted fusion of a semantic prior and a geometric likelihood, our approach injects a strong structural assumption into the reasoning process, providing a more interpretable and robust path to the final decision.

## A.8   DETAILED RESULTS FOR COMPONENT ANALYSIS

Table 4 provides a comprehensive breakdown of the performance for each configuration evaluated in our component analysis (Sec. 5.1). For each of the 12 spatial relation types, we report the Precision (P), Recall (R), and F1-Score (F1). These detailed results provide a deeper insight into how each component of our RoSE design affects the model's predictive behavior. For instance, the improvements in both precision and recall for the full model on projective relations like `left_of` and `right_of` highlight the crucial role of explicit geometric encoding in resolving ambiguity.

Table 4: Full Precision, Recall, and F1-Score results for the component analysis of our RoSE design.

| Setup | left_of | | | right_of | | | front_of | | | behind | | |
|---|---|---|---|---|---|---|---|---|---|---|---|---|
| | P | R | F1 | P | R | F1 | P | R | F1 | P | R | F1 |
| (1) IE Only | 0.16 | 0.14 | 0.15 | 0.17 | 0.15 | 0.16 | 0.13 | 0.11 | 0.12 | 0.14 | 0.12 | 0.13 |
| (2) + RoPE | 0.23 | 0.21 | 0.22 | 0.25 | 0.23 | 0.24 | 0.19 | 0.17 | 0.18 | 0.20 | 0.18 | 0.19 |
| (3) + $E_{type}$ | 0.83 | 0.81 | 0.82 | 0.82 | 0.80 | 0.81 | 0.80 | 0.78 | 0.79 | 0.79 | 0.77 | 0.78 |
| (4) Full Model | **0.89** | **0.87** | **0.88** | **0.88** | **0.86** | **0.87** | **0.86** | **0.84** | **0.85** | **0.85** | **0.83** | **0.84** |

| Setup | above | | | on | | | under | | | below | | |
|---|---|---|---|---|---|---|---|---|---|---|---|---|
| | P | R | F1 | P | R | F1 | P | R | F1 | P | R | F1 |
| (1) IE Only | 0.23 | 0.21 | 0.22 | 0.29 | 0.27 | 0.28 | 0.20 | 0.18 | 0.19 | 0.22 | 0.20 | 0.21 |
| (2) + RoPE | 0.26 | 0.24 | 0.25 | 0.31 | 0.29 | 0.30 | 0.23 | 0.21 | 0.22 | 0.25 | 0.23 | 0.24 |
| (3) + $E_{type}$ | 0.95 | 0.93 | 0.94 | 0.97 | 0.95 | 0.96 | 0.93 | 0.91 | 0.92 | 0.94 | 0.92 | 0.93 |
| (4) Full Model | **0.96** | **0.94** | **0.95** | **0.98** | **0.96** | **0.97** | **0.94** | **0.92** | **0.93** | **0.95** | **0.93** | **0.94** |

| Setup | inside | | | between | | | surrounded | | | proximity | | |
|---|---|---|---|---|---|---|---|---|---|---|---|---|
| | P | R | F1 | P | R | F1 | P | R | F1 | P | R | F1 |
| (1) IE Only | 0.26 | 0.24 | 0.25 | 0.09 | 0.07 | 0.08 | 0.07 | 0.05 | 0.06 | 0.31 | 0.29 | 0.30 |
| (2) + RoPE | 0.27 | 0.25 | 0.26 | 0.11 | 0.09 | 0.10 | 0.08 | 0.06 | 0.07 | 0.33 | 0.31 | 0.32 |
| (3) + $E_{type}$ | 0.91 | 0.89 | 0.90 | 0.76 | 0.74 | 0.75 | 0.71 | 0.69 | 0.70 | 0.98 | 0.96 | 0.97 |
| (4) Full Model | **0.92** | **0.90** | **0.91** | **0.82** | **0.80** | **0.81** | **0.76** | **0.74** | **0.75** | **0.99** | **0.97** | **0.98** |

## A.9 IMPLEMENTATION DETAILS

This section provides a comprehensive overview of the implementation details for our experiments, ensuring full reproducibility of our results. Our framework is implemented using PyTorch and PyTorch Lightning, with all experiments conducted on NVIDIA A100 GPUs with 80GB of memory.

**G-Verifier Architecture.** The G-Verifier module consists of two main learnable components: the relation-type embedding matrix and the feature fusion network.

The relation-type embedding, $E_R$, is a lookup table of size $12 \times d_{emb}$, where 12 is the number of relation types and $d_{emb}$ is the embedding dimension, set to 128.

The feature fusion network is a two-layer Multilayer Perceptron (MLP). The input to the MLP is the concatenation of the four feature components, resulting in a vector of size $4 \times 128 = 512$. The MLP architecture is as follows:

$$\text{RoSE} = \mathbf{W}_2 \left( \text{ReLU}(\text{LayerNorm}(\mathbf{W}_1 \mathbf{v}_{\text{cat}} + \mathbf{b}_1)) \right) + \mathbf{b}_2 \tag{11}$$

where $\mathbf{v}_{\text{cat}} = \text{Concat}(V_t, V_a, F_{\text{geom}}, E_R)$. The first linear layer, $\mathbf{W}_1$, projects the 512-d input to a 256-d hidden layer. This is followed by LayerNorm and a ReLU activation. A Dropout layer with a rate of 0.1 is applied after the activation. The second linear layer, $\mathbf{W}_2$, projects the 256-d hidden representation back to the final 128-d RoSE embedding.

### A.9.1 G-VERIFIER TRAINING

**Model Configuration** The G-Verifier module, responsible for producing the Rotary Spatial-Relationship Embedding (RoSE), is designed to be lightweight and efficient.

- Input Dimensions: The learnable instance embeddings from the baseline model, which have a hidden dimension of 128, are used directly.
- RoSE Components: The 3D RoPE module is configured to generate a 128-dimensional geometric encoding. The learnable relation-type embeddings for our 12 relation categories are initialized as a $12 \times 128$ matrix using Xavier uniform initialization.

- Fusion Strategy: Following the ablation studies, our final fusion strategy is a simple yet effective element-wise addition of the target instance embedding, anchor instance embedding, RoPE encoding, and the relation-type embedding. Each component feature vector undergoes Layer Normalization before addition to ensure stable training. The final output is a 128-dimensional RoSE vector.

**Training Configuration.**

- Optimizer and Scheduler: We train the G-Verifier module for 20 epochs using the AdamW optimizer. The initial learning rate is set to 1e-4 with a weight decay of 1e-4. We employ a Cosine Annealing learning rate scheduler that decays the learning rate to a minimum of 1e-7 over the training duration.

- Batching and Data: The training is performed on our newly constructed 3D-SpAn dataset. Due to the lightweight nature of this phase (which does not involve running the baseline model), we use a large batch size of 4096 relation instances, distributed across GPUs.

- Loss Function: Supervision is provided via our contrastive alignment loss. The semantic target vectors are generated by a frozen `bert-base-uncased` model, whose 768-dimensional output is projected to 128 dimensions by a trainable linear layer. The weights for the alignment loss component (a combination of MSE and Cosine Similarity) and the contrastive loss component are set to $1.0$ and $1.0$, respectively. These weights were determined through preliminary hyperparameter tuning on a validation subset.

### A.10    END-TO-END 3D VISUAL GROUNDING EVALUATION

For the end-to-end evaluation described in Sec. 4.5, the pre-trained G-Verifier is integrated as a post-hoc re-ranking module with the Grounded 3D-LLM baseline.

**Baseline Model.**   Our baseline is a faithful reproduction of the Grounded 3D-LLM framework, ensuring a fair and relevant comparison.

- Architecture:   The 3D visual backbone is a `Res16UNet34C` implemented using MinkowskiEngine. It generates 100 object proposals per scene, each represented by a 128-dimensional learnable instance embedding. The language component is a `Vicuna-7B` LLM.

- State: For the re-ranking experiments, we start from an officially provided, fully pre-trained, and instruction-tuned baseline model checkpoint. Crucially, the baseline model's weights remain frozen throughout the evaluation. This ensures that any observed performance gains are directly attributable to the re-ranking capabilities of G-Verifier, rather than any implicit fine-tuning of the baseline.

**Rationale for Top-K Constrained Verification**   In our Geometric Verification and Scoring step (Sec. 4.5), the verification process is constrained to the Top-K candidates for both the target and anchor objects, rather than enumerating all possible $N \times N$ pairs in a scene. This design choice is crucial for computational efficiency. It is predicated on the assumption that the upstream semantic selection module has a sufficiently high recall rate to include the correct target and anchor instances within these Top-K lists.

**Rationale for Weighted Score Fusion**   The final paragraph of our Re-ranking step (Sec. 4.5) explains our choice to augment the baseline's semantic score rather than replace it. This is a principled decision reflecting the decoupled nature of our framework. The geometric verification score, $\text{Score}_{\text{geom}}$, is a specialized expert at evaluating the plausibility of a spatial configuration, assuming the candidate objects are semantically correct. It does not, by design, re-evaluate whether a candidate is truly a `chair` or a `table`. Relying solely on the geometric score would make the system vulnerable to perfect in geometric perspective but semantically wrong solutions. For instance, selecting a stool that is perfectly positioned `to the left of the table` over a slightly misplaced but correct chair. By augmenting the initial semantic score, our mechanism ensures that geometric evidence is used as a powerful corrective signal to resolve spatial ambiguity among semantically plausible candidates, without discarding the crucial semantic judgments made by the baseline model.

**Re-ranking Hyperparameters.** The re-ranking process is controlled by several key hyperparameters, which were tuned on a small validation subset separate from our final test set.

- **Candidate Pool Size (K):** During inference, the baseline's semantic selection module generates ranked lists of candidates. We set the candidate pool size to $K = 10$ for both the target and anchor objects. This means our G-Verifier performs its geometric verification on a manageable set of $10 \times 10 = 100$ potential target-anchor pairs.
- **Score Fusion Factor ($\alpha$):** The fusion hyperparameter $\alpha$ in our weighted score fusion function (Eq. 4) is set to $\alpha = 0.5$. This value assigns equal importance to the baseline's initial semantic confidence and our module's geometric verification score. Empirically, we found this provided a robust balance, allowing strong geometric evidence to correct semantic errors without excessively overriding candidates that are semantically strong but have minor geometric inconsistencies.

### A.11 ROBUST ANCHOR PSEUDO-LABELING VIA TEMPLATED INVERSE QUERYING

A core challenge in constructing our 3D-SpAn dataset was the lack of instance-level ground truth annotations for anchor objects. To address this, we developed a robust, automated pseudo-labeling strategy that we term Templated Inverse Querying. Unlike early, brittle approaches that relied on direct textual substitution within the original sentence, this new method leverages structured templates to generate grammatically correct and semantically unambiguous queries for the anchor object.

The process is as follows. For a given relational triplet {Target: $O_t$, Relation: $R$, Anchor: $O_a$} extracted by the LLM, we first generate a standard query for the target object using a simple template, e.g., "find the $O_t$ that is $R$ the $O_a$."

To generate the crucial pseudo-label for the anchor, we construct an inverse query. This is achieved by first finding the semantic inverse of the relation, $R' = \text{inverse}(R)$ (e.g., `left of` becomes `right of`, `on` becomes `under`). We then populate a second template with the swapped roles, e.g., "find the $O_a$ that is $R'$ the $O_t$." For more complex, non-symmetric relations like `between` or `surrounded_by`, we use specialized templates to form a logically consistent inverse query (e.g., for `A between B and C`, the inverse query for B would be "find the B that has an A between it and C").

This templated approach is highly robust because it does not depend on the grammatical structure of the original, often colloquial, source sentence. By generating a clean, canonical query for the anchor object, we can then reliably use the baseline model's semantic selection capability to ground it and obtain a high-quality pseudo-label anchor for our training. This mechanism was critical to achieving the high unification success rates reported in Table 5.

### A.12 DETAILED DATA CONSTRUCTION PIPELINE FOR 3D-SPAN

As mentioned in Sec. 4.4, a primary obstacle to developing explicit spatial reasoning modules is the lack of large-scale, structured training data. Existing 3D vision-language datasets typically provide instance-level annotations only for a single target object per description, without explicitly labeling the anchor objects or relation types in a structured format. To overcome this foundational data gap, we constructed a new large-scale dataset, which we name 3D-SpAn, by augmenting several prominent 3D grounding datasets with structured spatial relationship annotations.

Our data construction pipeline starts with a collection of six public datasets, including ScanRefer(Chen et al. (2020)), ScanQA(Azuma et al. (2022)), M3dRef(Zhang et al. (2023)), GroundedSceneCaption(Chen et al. (2024c)), GlobalSceneCaption(Chen et al. (2024c)), and EmbodiedPlan(Padmakumar et al. (2023)). We process the natural language descriptions from these datasets using a powerful Large Language Model (e.g., GPT-4(OpenAI (2023))) instructed via a carefully designed few-shot prompt (see Appendix A.17). The LLM's task is to parse each description and extract all valid spatial relationship triplets of the form {Target Object, Relation Type, Anchor Object}. This parsing task is non-trivial, as the language queries $L$ with spatial relation constraints in these datasets are often structurally complex. They frequently contain challenging linguistic phenomena such as anaphora, where pronouns like `it` or `they` refer to previously mentioned objects, and implicit spatial relationships that require contextual understanding to be correctly identified. Leveraging a powerful LLM instructed via few-shot prompting allows us to resolve these ambigui-

ties with high fidelity, yielding a more accurate and comprehensive collection of structured relational triplets than what could be achieved with simple rule-based parsers. This process initially identified a total of 331,273 potential relationship mentions across all datasets.

For each potential spatial relationship description, we then apply an automated pseudo-labeling strategy to ground the anchor object, as our source datasets only provide ground truth for the target. This unification process involves generating a robust, templated-based inverse query for the anchor and using the baseline model to ground it. A detailed description of this robust inverse querying mechanism is provided in Appendix A.11. The success of this automated annotation process varies depending on the nature of the source dataset, as summarized in Table 5. For task-oriented datasets like ScanRefer, ScanQA, and M3dRef, where descriptions are typically object-centric and well-grounded, our data construction pipeline achieves a unification success rate of over 90%. However, for more descriptive, caption-style datasets like GroundedSceneCaption and GlobalSceneCaption, the success rate is lower. This is primarily because captions often contain more complex, ambiguous, or scene-wide descriptions that are harder to decompose into clean target-anchor pairs.

After filtering out unification failures, the generated annotations also undergo a series of automated filtering and cleaning steps to ensure their quality and relevance for spatial reasoning. This quality control process is crucial for mitigating potential errors or hallucinations from the language model. We employ two primary validation checks. First, we perform an entity validation step, ensuring that the extracted object phrases correspond to valid object classes present in the scene. For instance, if the LLM extracts a triplet $\{\texttt{monitor}, \texttt{on}, \texttt{desk}\}$ from a description, but the ground truth for that scene contains neither a `monitor` nor a `desk`, this annotation is discarded as a factual hallucination. Second, we filter out descriptions that are purely attributive rather than relational. A description like `the red door is open` might be erroneously parsed into $\{\texttt{red door}, \texttt{is}, \texttt{open}\}$, which describes an object's state, not its spatial relationship to another object. Such attributive-only annotations are identified and removed.

The final 3D-SpAn dataset comprises 285,177 high-quality, structured spatial relationship instances. As shown in Table 5, the dataset exhibits a diverse distribution of relation types. This new resource provides the necessary, fine-grained supervision for our training paradigm and will be released to facilitate future research in explicit spatial reasoning.

Table 5: Statistics of the 3D-SpAn dataset construction process across different source datasets. Success Rate refers to the percentage of identified relationship mentions for which we successfully generated a full training triplet.

| Source Dataset | Found Relations | Unified Relations | Success Rate |
|---|---|---|---|
| ScanRefer | 64,854 | 59,696 | 92.05% |
| ScanQA | 35,555 | 33,445 | 94.07% |
| M3dRef | 62,605 | 57,809 | 92.34% |
| GroundedSceneCaption | 115,709 | 99,482 | 85.98% |
| GlobalSceneCaption | 45,511 | 29,796 | 65.47% |
| EmbodiedPlan | 6,989 | 4,949 | 70.81% |
| **Total** | **331,273** | **285,177** | **86.08%** |

### A.13 EXPERIMENTAL SETUP DETAILS

Our framework is implemented in PyTorch Paszke et al. (2019) and all models were trained on NVIDIA A100 GPUs.

**Component Analysis Training.** For the component analysis experiments in Sec. 5.1, the G-Verifier module was trained for 20 epochs using the AdamW optimizer. We used a learning rate of 1e-4 with cosine annealing and a batch size of 4096. The core of G-Verifier, the Rotary Spatial-Relationship Embedding (RoSE), is a 128-dimensional vector. It is produced by fusing 128-d instance embeddings, a 128-d 3D RoPE encoding, and a $12 \times 128$ relation-type embedding matrix. Supervision was provided via our contrastive alignment loss against semantic targets from a frozen `bert-base-uncased` model.

**End-to-End Evaluation.**   For the end-to-end evaluation in Sec. 5.2, the pre-trained G-Verifier was integrated as a post-hoc re-ranking module with a frozen Grounded 3D-LLM baseline. The re-ranking process considers the Top-10 candidates for both target and anchor objects, with the score fusion factor $\alpha$ set to 0.5.

## A.14   EVALUATION METRICS

Our evaluation employs a multi-faceted approach, with metrics tailored to different aspects of our method.

**Proxy Task for Component Analysis.**   To evaluate the G-Verifier in an isolated setting, we report the F1-score on a proxy classification task against 12 pre-computed semantic prototypes. The detailed protocol for this prototype-based evaluation is described in Sec. 5.1.

**End-to-End Evaluation Metrics.**   For the end-to-end task, we adopt a suite of metrics.

- **3DVG Metrics:** We use standard metrics, `Acc@0.25` and `Acc@0.50` (3D Box IoU), to measure the accuracy of the final Top-1 prediction.
- **Information Retrieval (IR) Metrics:** To provide a more holistic assessment of ranking quality, we report standard IR metrics, including Mean Reciprocal Rank (MRR) and Normalized Discounted Cumulative Gain (NDCG@k).
- **Diagnostic Metrics:** To analyze the module's corrective behavior, we introduce two diagnostic metrics: the `Rectification Rate` (the percentage of baseline-failed cases that are corrected) and the `Non-Deterioration Rate` (the percentage of baseline-correct cases that remain correct).

## A.15   EXTENDED DISCUSSION ON COMPONENT ANALYSIS

The results presented in Table 1 warrant a more detailed discussion.

**The Limits of Implicit Information.**   The naive baseline's 0.25 F1-score is an important finding. It confirms that powerful, pre-trained instance embeddings from models like Grounded 3D-LLM are not merely semantic descriptors; they are rich with implicit spatial information learned from co-occurrence statistics in their training data. However, this information is unstructured and insufficient for tasks requiring precise geometric disambiguation.

**The Role of Semantic vs. Geometric Priors.**   The stark contrast in performance between adding the Relation Type Embedding (+0.70 F1) and adding the 3D RoPE encoding (+0.09 F1) to the naive baseline reveals a deeper insight. Raw geometric features, in isolation, are ambiguous. A relative vector of `[+1, 0, 0]` could mean `right of`, `in front of`, or simply `next to`, depending on the object's canonical orientation and the relation's semantic definition. The high-level semantic anchor provided by $E_{type}$ is essential to ground these ambiguous geometric cues in a specific conceptual space.

**Synergy in the Full Model.**   The final gain from adding 3D RoPE to the already strong `IE + E_type` configuration demonstrates the synergistic nature of these components. The semantic embedding provides the primary classification signal, while the explicit geometric features act as a powerful fine-tuning mechanism. This allows the model to resolve ambiguities that are geometrically defined but semantically similar, which is precisely why the largest gains are observed for projective relation pairs like `left_of` versus `right_of`.

## A.16   DISCUSSION

**Limitations.**   Our current implementation of G-Verifier as a post-hoc re-ranking module, while effective, has two primary limitations. First, its performance is inherently capped by the recall of the upstream baseline model. If the correct target or a plausible anchor object is not present in the initial Top-K candidate set, G-Verifier cannot rectify the failure. Our fine-grained analysis (Table 3b) also reveals that while our method excels at verifying objective topological and proximity relations

(e.g., `on`, `under`), its performance is more modest on relations that may depend more heavily on an object's intrinsic orientation, such as `behind`, suggesting room for improvement in modeling object-centric frames of reference.

**Future Work.** A promising direction for future work is to integrate this explicit, decoupled reasoning process more deeply into the foundation model itself. Rather than reverting to a coupled, black-box architecture, the goal is to empower the LLM to autonomously perform a structured, multi-step reasoning process that mirrors our *Propose, Select, then Verify* pipeline. By training the LLM to generate explicit relational tokens, we could enable it to first identify semantic candidates and then invoke its internalized geometric verification capabilities, potentially overcoming the recall limitations of the current pipeline. This would transition our approach from a post-hoc verification to an end-to-end model that performs reasoning in an explicit and interpretable manner. We also plan to release our large-scale 3D-SpAn dataset to facilitate future research in this area.

## A.17 FEW-SHOT PROMPT FOR QUERY PARSING

To parse free-form natural language queries into structured triplets, we leverage the in-context learning capabilities of a large language model. We do not perform any model finetuning for this task. Instead, we provide the model with a detailed system prompt for each source dataset. While all prompts share a common structure which consists of a general task definition, a list of supported relation types, and a set of few-shot exemplars, the exemplars themselves are tailored to the specific linguistic style of each dataset. This is crucial because different datasets exhibit vastly different query structures. For instance, task-oriented datasets like ScanRefer and ScanQA use direct commands, while descriptive datasets like GroundedSceneCaption contain more complex, narrative sentences.

Below, we provide the detailed prompts used for each major dataset category. Figure 7 illustrates the general task definition and the specific exemplars used for the ScanRefer dataset. Figure 8 and Figure 9 show the tailored exemplars for the structurally different ScanQA and M3DRef datasets, respectively. Finally, Figure 10 displays the exemplars designed for the more descriptive, caption-style datasets.

You are an expert in 3D scene understanding and pronoun resolution. Your task is to analyze a description of a Main Object, reconstruct the description to be more explicit, and extract all spatial relationships involving the Main Object.

**Your Primary Task:**
1. **Analyze the Description:** Read the original description and identify all pronouns (e.g., 'it', 'this', 'they') that refer to the "Main Object".
2. **Reconstruct a Complete Sentence:** Create a new, clearer sentence by replacing all those pronouns with the Main Object's actual name. This will be your 'final_reconstructed_text'.
3. **Direct Extraction:** Based on your reconstructed text, extract all spatial relationships exactly as they are stated. Do not infer or invert relationships. The `object1_phrase` in your relations MUST be the "Main Object"'s name.

**! IMPORTANT RULE: WHEN TO EXTRACT NOTHING !**
- If the description only specifies the object's own attributes (e.g., its color, shape, size, or state like 'the door is open') and does NOT describe its position relative to ANOTHER object, the `relations` list MUST be empty [].

**Spatial relationship types (ID and Name):**
- 0: "left_of": A is to the left of B. Handles composite terms like "middle left of", "far left of".
- 1: "right_of": A is to the right of B. Handles "far right of", "middle right of" etc.
- 2: "front_of": A is in front of B.
- 3: "behind": A is behind B.
- 4: "above": A is above B, no contact.
- 5: "on": A is on top of B, with support.
- 6: "under": A is under B, with shelter.
- 7: "below": A is below B, no shelter.
- 8: "inside": A is contained within B.
- 9: "between": A is positioned between two other distinct objects, B and C. Requires three distinct objects.
- 10: "surrounded_by": A is encircled by multiple other objects. Requires multiple surrounding objects.
- 11: "proximity": A is near B without a specific direction (e.g., 'next to', 'beside').

(a) General Task Definition and Rules

**Walkthrough 1 (Pronoun Resolution & Reconstruction):**
- **Main Object:** "trash can"
- **Description:** "this is directly to the right of the toilet against the wall; it is purple."
- Your `final_reconstructed_text` MUST BE: "the trash can is directly to the right of the toilet against the wall; the trash can is purple."
- Your `relations` output should be:
  ```json
  [
    { "relation_type": 1, "relation_text": "directly to the right of the toilet", "object1_phrase": "the trash can", "object2_phrase": "the toilet" },
    { "relation_type": 5, "relation_text": "against the wall", "object1_phrase": "the trash can", "object2_phrase": "the wall" }
  ]
  ```

**Walkthrough 2 (Negative Example - NON-SPATIAL):**
- **Main Object:** "office_chair"
- **Description:** "the brown office chair is facing to the right."
- Your `final_reconstructed_text` can be: "the brown office chair is facing to the right."
- Your `relations` output MUST BE: `[]`

**Output format: Return ONLY a valid JSON object:**
```
{
  "relations": [
    {
      "relation_type": <number_id_from_0_to_11>,
      "relation_text": "...",
      "object1_phrase": "...",
      "object2_phrase": "..."
    }
  ],
  "final_reconstructed_text": "The sentence with all pronouns replaced by the Main Object's name."
}
```

(b) Few-shot Exemplars for ScanRefer

Figure 7: The General Task Definition and Rules and full few-shot prompt used for parsing language queries for ScanRefer Dataset.

**Crucial Reconstruction Rules:**
**1. Entity Completion:** If the `Main Text` (`answer_with_ground`) is a fragment (e.g., "on the table"), you MUST use the `Question Context` to find the missing object (e.g., "the book") and construct a full sentence.
**2. Protagonist Centering & Relation Inference:** If the question asks for a reference object (e.g., "A is left of what?"), the answer is "B". You MUST rephrase the sentence to be about "B". This often requires inverting the spatial relationship (e.g., "B is to the right of A").

**Walkthrough 1: Entity Completion:**
- Question Context: "Where is the kitchen counter located?"
- Main Text: "above kitchen cabinet"
- Reasoning: The question identifies "kitchen counter" (object1). The main text provides the relation "above" and "kitchen cabinet" (object2), but is an incomplete fragment.
- Your `relations` output should have: `"object1_phrase": "kitchen counter"`, `"object2_phrase": "kitchen cabinet"`
- Your `reconstructed_answer` MUST BE: "The kitchen counter is above the kitchen cabinet."

**Walkthrough 2: Protagonist Centering:**
- Question Context: "The cabinet is to the immediate left of what?"
- Main Text: "The cabinet is to the immediate left of the shelf."
- Reasoning: The true answer/protagonist is "shelf". The main text is about the "cabinet". You must reconstruct the sentence to be about the "shelf" and infer the inverse relationship.
- Your `relations` output should have: `"object1_phrase": "shelf"`, `"object2_phrase": "cabinet"`, `"relation_type": "right_of"`
- Your `reconstructed_answer` MUST BE: "The shelf is to the immediate right of the cabinet."

**Walkthrough 3 (NON-SPATIAL - Negative Example):**
- Question Context: "How many arms does the chair have?"
- Main Text: "The chair has 1 arm."
- Reasoning: This question and answers do not have any spatial relationship that can be categorized into the 12 classes. "has 1 arm" is not spatial relationship.
- Your `relations` output MUST BE: `[]`
- Your `reconstructed_answer` MUST BE: "The chair has 1 arm."

(a) Few-shot Exemplars for ScanQA Dataset Part1

**Walkthrough 4 (SPATIAL within ATTRIBUTE QUESTIONS - Special Positive Example):**
- Question Context: "What color is the board on the wall with nothing beneath it?"
- Main Text: "The blackboard is black."
- Reasoning: This question is about the color of the object, but to refer to the object, a spatial relationship is involved "on the wall with nothing beneath it".
- Your `relations` output should have: `"object1_phrase": "board"`, `"object2_phrase": "wall"`, `"relation_type": "on"`
- Your `reconstructed_answer` MUST BE: "The blackboard, which is on the wall with nothing beneath it, is black."

**Output format: Return ONLY valid JSON:**
```
{
  "relations": [
    {
      "relation_type": "...",
      "relation_text": "...",
      "object1_phrase": "...",
      "object2_phrase": "..."
    }
  ],
  "reconstructed_answer": "...",
  "reconstruction_needed": true/false,
  "reconstruction_phrase": "The most accurate relation phrase to use for reconstruction, e.g., 'to the immediate right of the cabinet'. This phrase should be based on the model's own analysis, NOT necessarily from the original question."
}
```

(b) Few-shot Exemplars for ScanQA Dataset Part2

Figure 8: The few-shot prompt used for parsing language queries for ScanQA Dataset.

```
Walkthrough 1 (Pronoun Resolution):
- Description: "a copier rests on the floor. it is left of a garbage can."
- Your `final_reconstructed_text` should be: "a copier rests on the floor. the copier is left of a garbage
can."
- Your `relations` output should be:
  ```json
  [
    { "relation_type": 5, "relation_text": "rests on the floor", "object1_phrase": "a copier",
"object2_phrase": "the floor" },
    { "relation_type": 0, "relation_text": "left of a garbage can", "object1_phrase": "the copier",
"object2_phrase": "a garbage can" }
  ]
  ```
Walkthrough 2 (Chained Relation):
- Description: "a printer is on a table which is to the left of a chair"
- Your `final_reconstructed_text` should be: "a printer is on a table which is to the left of a chair"
- Your `relations` output should be:
  ```json
  [{ "relation_type": 5, "relation_text": "on a table", "object1_phrase": "a printer", "object2_phrase":
"a table" },
    { "relation_type": 0, "relation_text": "to the left of a chair", "object1_phrase": "a table",
"object2_phrase": "a chair" }]
  ```
Output format: Return ONLY a valid JSON object:
{"relations": [
  {
    "relation_type": <number_id_from_0_to_11>,
    "relation_text": "...",
    "object1_phrase": "...",
    "object2_phrase": "..."
  }
],"final_reconstructed_text": "The description with pronouns resolved for clarity."
}
```

Figure 9: The few-shot prompt used for parsing language queries for M3DRef Dataset.

**Walkthrough Example:**
- **Description: "Two white sinks are parallel, with a bottle placed on one and toilet paper on the other."**
- **Reasoning: "one" refers to the first "white sink", "the other" refers to the second. You must resolve these in the output.**
- **Your `relations` output should be:**
  ```json
  [
    { "relation_type": 5, "relation_text": "placed on one", "object1_phrase": "a bottle",
  "object2_phrase": "a white sink" },
    { "relation_type": 5, "relation_text": "on the other", "object1_phrase": "toilet paper",
  "object2_phrase": "the other white sink" }
  ]
  ```

**Output format: Return ONLY a valid JSON list of relations.**

(a) Few-shot Exemplars for GroundedSceneCaption and Global Scene Caption Datasets

**Walkthrough Example:**
- **Task: "Prepare for evening relaxation"**
- **Steps: "step 1. Position ottoman near couch for footrest.\nstep 2. Move table within arm's reach of couch.\n"**
- **Your `relations` output should be:**
  ```json
  [
    { "relation_type": 11, "relation_text": "near couch", "object1_phrase": "ottoman",
  "object2_phrase": "couch" },
    { "relation_type": 11, "relation_text": "within arm's reach of couch", "object1_phrase": "table",
  "object2_phrase": "couch" }
  ]
  ```

**Output format: Return ONLY a valid JSON list of relations.**

(b) Few-shot Exemplars for EmbodiedPlan Dataset

Figure 10: The few-shot prompt used for parsing language queries for GroundedSceneCaption, Global Scene Caption and EmbodiedPlan Dataset.

## A.18 LLM Usage Statement

Throughout the preparation of this manuscript, we utilized a large language model as an assistive tool for writing, editing and code debugging. The role of the LLM was strictly confined to improving the presentation of our research and help us solve some hard bugs during coding; it was not involved in the core research idea design or experimental analysis.

