# OpenReview forum: "G-Verifier: Geometric Verifier for Robust 3D Point Cloud Semantic Search with Spatial Relation Reasoning"
_ICLR.cc/2026/Conference — Submitted to ICLR 2026_

### Official Review · Reviewer_GRtD · 2025-10-19

**Soundness:** 2
**Presentation:** 2
**Contribution:** 1
**Rating:** 2
**Confidence:** 4

**Summary:**

This paper introduces G-Verifier, a module to encode explicit spatial relations between objects to re-rank the object proposals (candidates) in 3DVG. G-Verifier outputs RoSE features for each candidate and applies contrastive learning to implement the re-ranking. This paper also introduces 3D-SpAn, a large-scale 3DVG dataset with structured explicit spatial relationship annotations, to train and evaluate G-Verifier.  Experimental results show that adding G-Verifier improves the backbone model (Grounded 3D LLM [a]).

a. [Grounded 3D-LLM with Referent Tokens](https://arxiv.org/abs/2405.10370)

**Strengths:**

1. This paper is well-structured.
2. Body texts are easy to follow.
3. The proposed 3D-SpAn dataset adds value to the 3DVG community.
4. Disentangling semantics and spatial relations processing can be a good attempt.

**Weaknesses:**

1. Many typos, missing spaces, and incorrect citation formats, such as "F1-score(0.96)" in line 028, "...geometry. (Xu et al. (2024b))." in line 070, and "(MLP) Haykin (1994) to" in line 302.

2. Over-squeezed spaces between images and body text. Figure 1 & 2 abuse \vspace.

3. In line 157, the authors misrefer to Figure 6 in the appendix, and Figures 2 & 6 are the same.

4. Vigor [a] and CoT3DRef [b] should be discussed because they have been attempts to decouple the semantic and geometric information of 3DVG, which is very relevant to your core argument.

5. G-Verifier and the proposed 3D-SpAn dataset seem to target a simplified scenario where object relations are a pre-defined closed set, like SR3D [c]. However, real-world scenarios may be users verbalizing natural, lengthy descriptions without fixed templates, as NR3D [c]. It is unclear to me how G-Verifier will tackle such natural descriptions.

6. G-Verifier is a second-stage refinement on another model's outputs—Grounded 3D-LLM itself has had the ability to propose object candidates and select the most possible one. That is, G-Verifier is a "plug-in" to backbone 3DVG models, analogous to CoT3DRef. In this case, G-Verifier should be evaluated on several backbone models to verify its general benefits. The only experiment on performance improvement is on Grounded 3D-LLM [d] (Table 2), where numbers seem not significant, given that the extra stage and parameters are used.

7. The novelty (settings, engineering techniques, and insights) of G-Verifier is limited. G-Verifier merges explicit spatial info into the last process, while the main reason why previous works did implicit learning is because they consider scenarios where spatial info is complex and descriptions have open forms.

a. [Data-Efficient 3D Visual Grounding via Order-Aware Referring](https://arxiv.org/pdf/2403.16539)
b. [Chain-of-Thoughts Data-Efficient 3D Visual Grounding](https://arxiv.org/abs/2310.06214)
c. [ReferIt3D: Neural Listeners for Fine-Grained 3D Object Identification in Real-World Scenes](https://github.com/referit3d/referit3d)
d. [Grounded 3D-LLM with Referent Tokens](https://arxiv.org/abs/2405.10370)

**Questions:**

I may raise my score if authors address the above weaknesses. My major concerns include limited novelty, simplified settings, and insufficient experiments.

---

### Official Review · Reviewer_6Rhx · 2025-10-26

**Soundness:** 2
**Presentation:** 2
**Contribution:** 1
**Rating:** 2
**Confidence:** 4

**Summary:**

The paper proposes G-Verifier, a post-hoc geometric verification module intended to improve spatial reasoning in 3D visual grounding. The design follows a Propose, Select, Verify pipeline, where the final verification stage re-ranks candidate objects using a newly proposed RoSE relational embedding. Experiments show moderate improvements on a constructed benchmark.

**Strengths:**

- The paper is clearly written and motivated by well-known weaknesses of current 3DVG/3D-LLM models.
- The re-ranking idea is conceptually intuitive and easy to integrate into existing systems.
- The new dataset appears reasonably large and might be useful to the community.

**Weaknesses:**

- **Lack of Novelty**: The core idea, introducing a post-hoc re-ranking module using relation embeddings, is well-established in both 2D grounding and classical IR-style pipelines. Most components (contrastive alignment, RoPE encoding, relation-type embeddings, weighted score fusion) are direct adaptations of existing methods. The paper does not demonstrate what is fundamentally new beyond recombining known elements. If the authors present the dataset contribution as their primary contribution, they should emphasize the associated analysis and insights instead of stressing methodological novelty.
- **Confusing Experimental Design**: A critical issue is that the paper does not include comparisons against any existing 3D visual grounding or spatial reasoning baselines. The experiments only evaluate variants of the proposed method on a custom dataset, which makes it impossible to assess contribution or practical significance.
- **Marginal Practical Impact**:  The reported improvements are small (e.g., +2.5% Acc@0.50) and limited to a curated evaluation split rather than standard benchmarks. There is no evidence that the proposed verifier improves generalization in broader or real-world conditions.
- **Insufficient Detail in Data**: The model relies heavily on the newly constructed 3D-SpAn dataset. However: The pseudo-labeling process is under-specified. The dataset is not benchmarked against alternatives. It is unclear whether the module performs well without this dataset, limiting reproducibility and applicability.
- **Limited Insight Into Failure Cases**: The analysis focuses on overall performance but does not sufficiently explore when and why the verifier fails. Certain relation types show no improvement (e.g., behind, right of), suggesting that the approach may not truly address viewpoint-dependent spatial reasoning, despite claiming to.
- **Conceptual Gap in the Claimed Contribution**: The paper’s thesis is that decoupling semantic and geometric reasoning is necessary. However, the verifier still depends on semantic embeddings and does not provide insight into how “geometric reasoning” is being meaningfully separated rather than just appended.

**Questions:**

Please see Weaknesses

---

### Official Review · Reviewer_wVNo · 2025-10-27

**Soundness:** 3
**Presentation:** 3
**Contribution:** 2
**Rating:** 4
**Confidence:** 3

**Summary:**

This paper introduces G-Verifier, a geometric verifier and re-ranker designed for post-processing grounding results. To train this module, the authors constructed a large-scale 3D spatial relation annotation dataset, 3D-SpAn, and employed a specific language alignment training strategy. Experimental results indicate that the G-Verifier module possesses discriminative capabilities. When integrated into the end-to-end pipeline, it significantly enhances the baseline model's localization accuracy while maintaining high stability. However, as a post-processing module, the paper lacks sufficient experiments demonstrating the method's generality and efficiency.

**Strengths:**

1. This paper designed a post-hoc re-ranker, G-Verifier, which can be conveniently integrated into existing 3DVG frameworks.

2. The paper construct the large-scale 3D-SpAn dataset, featuring structured spatial relation annotations, holds significant value for relevant research within the community.

3. The results in the paper clearly demonstrate G-Verifier's effectiveness in enhancing spatial relation reasoning robustness, even when compared against a strong baseline model.

**Weaknesses:**

1. The paper primarily compares against the authors' re-implementation of the Grounded 3D-LLM baseline. While strong, it lacks direct comparison with the results reported in the original Grounded 3D-LLM paper or other recent SOTA 3DVG methods that also focus on spatial relations.
2.  G-Verifier functions as a post-processing step, introducing additional computation. The paper does not explicitly analyze the resulting increase in inference time or computational resource requirements.
3.  The authors acknowledge that using "inverse querying" to generate anchor pseudo-labels may introduce noise. Further discussion or quantification of this noise's impact on model training, final performance, and the model's robustness to such noise is recommended.
4.  The rationale behind the chosen feature fusion method within ROSE could be more clearly articulated. Additionally, the specific advantages of 3D RoPE compared to alternative 3D relative position encoding methods could be further elaborated.
5.  Does the method merely overfit to the authors' proposed dataset? Can this post-processing approach improve the model's general grounding capabilities?

Although this is a post-processing method, its overall simplicity and seamless integration with other methods are commendable. Therefore, if the authors can address these concerns, I will consider increasing my score.

**Questions:**

see in weakness

---

### Meta-Review · Area_Chair_nUDm · 2026-01-11

**Summary:**

This paper proposes G-Verifier, a post-hoc geometric verification / re-ranking module to improve 3D visual grounding, and introduces a new 3D-SpAn dataset with structured 3D spatial-relation annotations. Reviewers’ main concerns were limited novelty, insufficient comparisons to existing SOTA / standard benchmarks, and unclear generality beyond the authors’ specific setup. Additional concerns included added inference cost without efficiency analysis, and possible over-reliance on the new dataset / pseudo-label noise. These issues drove the decision discussion despite acknowledging the dataset’s potential value.

Overall: Even if the rebuttal improved clarity and added some extra experiments, the most load-bearing issues remain generality / baselines / standard benchmarks / novelty, especially given two strong rejects (2,2) vs. a single borderline (4).

**Reviewer Concerns:**

Reviewer Concerns* (no rebuttal)

Since there was no rebuttal/discussion, all concerns remain outstanding:

wVNo (4): missing broader SOTA comparisons, efficiency/latency analysis, pseudo-label noise impact, generality beyond 3D-SpAn.

6Rhx (2): limited novelty, weak/insufficient experimental design, marginal gains, lack of standard benchmarks and failure analysis.

GRtD (2): limited novelty, simplified setting, missing related work (Vigor/CoT3DRef), single-backbone evaluation, and presentation issues.

**Reviewer Scores:**

With no rebuttal/discussion, scores would likely remain unchanged.

---

### Decision · Program_Chairs · 2026-01-26

Reject